# Mass Spectrometry-Based Proteomic Analysis of Potential Host Proteins Interacting with N in PRRSV-Infected PAMs

**DOI:** 10.3390/ijms25137219

**Published:** 2024-06-29

**Authors:** Shijie Zhao, Fahao Li, Wen Li, Mengxiang Wang, Yueshuai Wang, Yina Zhang, Pingan Xia, Jing Chen

**Affiliations:** 1College of Veterinary Medicine, Henan Agricultural University, Longzi Lake 15#, Zhengzhou 450046, China; zsj1002935527@163.com (S.Z.); 15514847056@163.com (F.L.); lw839466399@163.com (W.L.); wmxjya@163.com (M.W.); 19515541370@163.com (Y.W.); yinazhang2020@henau.edu.cn (Y.Z.); 2College of Life Science, Henan Agricultural University, Longzi Lake 15#, Zhengzhou 450046, China

**Keywords:** PRRSV-N, LC–MS/MS, PPI, innate immunity

## Abstract

One of the most significant diseases in the swine business, porcine reproductive and respiratory syndrome virus (PRRSV) causes respiratory problems in piglets and reproductive failure in sows. The PRRSV nucleocapsid (N) protein is essential for the virus’ assembly, replication, and immune evasion. Stages in the viral replication cycle can be impacted by interactions between the PRRSV nucleocapsid protein and the host protein components. Therefore, it is of great significance to explore the interaction between the PRRSV nucleocapsid protein and the host. Nevertheless, no information has been published on the network of interactions between the nucleocapsid protein and the host proteins in primary porcine alveolar macrophages (PAMs). In this study, 349 host proteins interacting with nucleocapsid protein were screened in the PRRSV-infected PAMs through a liquid chromatography–tandem mass spectrometry (LC–MS/MS)-based proteomics approach. Bioinformatics analysis, which included gene ontology annotation, Kyoto Encyclopedia of Genes and Genomes database enrichment, and a protein–protein interaction (PPI) network, revealed that the host proteins interacting with PRRSV-N may be involved in protein binding, DNA transcription, metabolism, and innate immune responses. This study confirmed the interaction between the nucleocapsid protein and the natural immune-related proteins. Ultimately, our findings suggest that the nucleocapsid protein plays a pivotal role in facilitating immune evasion during a PRRSV infection. This study contributes to enhancing our understanding of the role played by the nucleocapsid protein in viral pathogenesis and virus–host interaction, thereby offering novel insights for the prevention and control of PRRS as well as the development of vaccines.

## 1. Introduction

The extremely infectious illness known as porcine reproductive and respiratory syndrome (PRRS) is brought on by PRRSV. The main characteristics are miscarriage, stillbirth of sows, and dyspnea of pigs at different ages [1]. One of the most significant infectious illnesses threatening the global pig industry’s growth is PRRS, which has resulted in enormous annual economic losses for China [2,3]. A PRRSV infection can induce low levels of neutralizing antibodies, and weak specific cellular immunity and innate immunity, leading to immunosuppression and persistent viral infection in pigs, which can then be complicated with multiple diseases [4]. The current mainstream method for preventing PRRSV is vaccination; however, the effectiveness of most vaccines remains limited. Therefore, further identification of host interactions during a PRRSV infection will help to understand the biological processes and therapeutic mechanisms of such an infection.

PRRSV is a positive-sense, enveloped, single-stranded RNA virus that is a member of the *Arteriviridae* family, order *Nidovirales*, and genus Arterivirus [5,6]. PRRSV isolates can be classified into two separate groups based on their genetic diversity, namely PRRSV-1 (formerly known as the European genotype) and PRRSV-2 (previously known as the North American genotype), whose genome sequences differ by roughly 40% [7]. The PRRSV genome spans 15.4 kb and has over eleven open reading frames (ORFs). It encodes eight structural proteins (GP2a, GP2b, GP3, GP4, GP5, GP5a, M, and N) and at least sixteen non-structural proteins (nsp1α, nsp1β, nsp2-6, nsp-2N, nsp-2TF, nsp7α, nsp7β, and nsp8-12) [8]. The nucleocapsid protein is the sole component of the viral capsid and interacts with itself in both covalent and non-covalent ways [9]. It is involved in genome replication and is also a key regulator of virus replication and virus–host interactions. The nucleocapsid protein regulates various processes, including ribosome biogenesis, inflammation, cell cycle progression, and IFN-β production [9,10,11]. As a result, the multifunctional nucleocapsid protein is an important target for researchers exploring the PRRSV infection mechanism.

As a result of the ongoing interactions between the virus and host during the co-evolution process, the host antiviral innate immune system has since changed, which has affected the virus’ capacity to manipulate the host defense system in order to spread [12]. There are previous reports about the interactions between host proteins and the PRRSV nucleocapsid protein in HEK293-T cells using GFP-traps and LC–MS/MS [13]. However, PAMs are the natural target cells for PRRSV infection. Thus, in the current study, we used a co-immunoprecipitation (Co-IP) combined with LC–MS/MS technique to identify 349 possible host proteins interacting with the PRRSV nucleocapsid protein in the infected PAMs, and then built a PPI network to provide a more complete understanding of the virus–host relationship. We also performed bioinformatics analysis of the N-interacting proteins. The PRRSV-N-interacting host proteins are associated with many physiological processes, including protein binding, DNA transcription, metabolism, and innate immune response, according to the results of the gene ontology annotation and pathway enrichment studies. We then demonstrated that the PRRSV nucleocapsid protein has interactions with the ATP-dependent RNA helicase (DDX1 and DDX3X), Sterile alpha motif and HD domain-containing protein 1 (SAMHD1), Poly(rC)-binding protein 2 (PCBP2), and Eukaryotic translation initiation factor 3 subunit G (EIF3G). Finally, we found that the nucleocapsid protein can promote the expression of the natural immune negative regulator PCBP2. As a result, the findings of this study will help to improve our understanding of the pathophysiology of PRRSV, providing new ideas and targets for the development of novel vaccines, as well as aid in PRRSV preventive and control efforts.

## 2. Results

### 2.1. Identification of Nucleocapsid Protein Interactions in the PRRSV-Infected PAMs by Liquid Chromatography Mass Spectrometry

The PAMs were infected with PRRSV, and samples were then collected at 12 and 48 h post-infection (hpi) to assess the expression of the nucleocapsid proteins. The expression level of the nucleocapsid protein progressively increased over the course of the PRRSV infection, reaching its peak at 36 hpi (Figure 1A). Subsequently, samples were collected specifically at 36 hpi for further interactome analysis. The cell lysates were immunoprecipitated using purified anti-PRRSV-N IgG at 36 hpi and subjected to silver staining to visualize the cellular proteins bound to the PRRSV nucleocapsid protein (Figure 1B). The PAMs in the control group were treated with RPMI Medium 1640 containing 2% fetal bovine serum, and their lysates were used to eliminate non-specific interactions. Distinct bands were observed, highlighting specific interactions compared to the negative control. Moreover, the differentially expressed bands between the N–IP and IgG–IP lanes were extracted from the gels and detected by LC– MS/MS. A total of 1029 proteins that interact with the nucleocapsid protein were identified. To decrease the possibility of incorrect peptide identification, only peptides with Mascot scores ≥ 30 were counted as identified. Each confident protein identification requires at least two unique peptides. By subtracting 680 ineligible binding proteins, 349 proteins were identified as potential N-interacting proteins in the PAMs. Appendix A contains the comprehensive details of the proteins that have been discovered.

### 2.2. Functional Analysis of Potential N-Interacting Proteins

To define the properties of the N-interacting proteins, GO (Gene Ontology) analysis was utilized to identify the potential functional pathways. All the genes have matching GO annotations. Figure 2 illustrates the three primary categories of GO analysis. Biological processes are usually complex processes in which a variety of molecular activities work together. The majority of N-interacting proteins were found to be involved in translation, protein transport, and protein stability (Figure 2A). Cellular components are cellular structures in which a gene product performs a specific function. Most N-interacting proteins functioned in cell components, cells, and organelles (Figure 2B). In general, molecular function refers to the actions that specific gene products can do. The majority of N-interacting proteins were shown to have binding, catalytic, and structural molecular activities (Figure 2C).

### 2.3. KEGG Pathway Annotation of the Potential Proteins Interacting with Nucleocapsid Protein

We performed pathway enrichment analysis using KEGG to better understand and predict the biological pathways of the N-interacting host protein candidates targeted by PRRSV. The top 20 enriched pathways with the greatest representation of each term were enumerated. Furthermore, the bulk of the target proteins were involved in genetic information processing, cellular processes, metabolism, and so on. In addition, the N-interacting proteins were predominantly enriched in metabolic pathways, ribosome, spliceosome, proteasome, and protein processing in the endoplasmic reticulum, antigen processing and presentation, and nucleocytoplasmic transport pathways (Figure 3).

### 2.4. Protein–Protein Interaction (PPI) Network Analysis of N-Interacting Proteins

In order to gain additional insight into the functional relationships between the N-interacting proteins and to identify specific functional complexes, a PPI network comprising 199 identified proteins (Figure 4) was built based on predictions made by the STRING search tool. With the stated number of nodes (199), the observed number of edges (210) for the network was much higher than the expected amount (108), suggesting that there are more interactions than one may have anticipated. Orthologous gene and protein clusters were identified using the MCL (Markov Cluster Algorithm) [14]. Different colors represent different clusters, and there are strong correlations between proteins in the same cluster. This enrichment suggests that different proteins in the same cluster are highly likely to work together and perform their biological functions.

### 2.5. Validation of the Proteins That Interact with the Nucleocapsid Protein by Co-IP

To further validate the protein interactions obtained from mass spectrometry, we conducted Co-IP. We selected innate immune-related proteins as proteins of interest to verify their interaction with the nucleocapsid protein. DDX1 and DDX3X play a central role in host IFN response [15]. As a constituent of the viral RNA sensor complex, DDX1 interacts with dsRNA and, in collaboration with DHX36, DDX21, and the adaptor protein TRIF, initiates the IFN-β response [16]. To enhance IFN-β production, DDX3X associates with viral dsRNA and recruits RIG-I stimulating factor IPS-1 [17]. In addition to their roles in innate immunity, both DDX1 and DDX3X are involved in various aspects of RNA metabolism, including protein translation, miRNA biosynthesis, mRNA synthesis and maturation as well as nuclear export [18]. Certain RNA viruses exploit these RNA helicases by utilizing their physical interactions to facilitate transcription and replication of the viral genome. SAMHD1 acts as a deoxynucleotide triphosphohydrolase (dNTPase). This enzyme controls the amounts of cytosolic dNTPs, which helps to limit the replication of viruses that rely on cellular dNTPs for genome replication [19,20]. SAMHD1 can suppress the NF-kB and IFN-I signaling pathways in response to pro-inflammatory stimuli and viral infections [21,22]. SAMHD1 interacts with inhibitors of nuclear factor kappa-B kinase epsilon (IKKε) and IFN regulatory factor 7 (IRF7), leading to IKKε inhibiting the phosphorylation of IRF7. Furthermore, SAMHD1 has been shown to inhibit the activity of IRF7-induced IFN-sensitive Response Element (ISRE) reporter factors [21]. EIF3G plays an important role in the translation of viral proteins. By targeting only the viral protein Rev, EIF3G mediates the action of cbp80/20 to suppress HIV-1 Gag synthesis [23]. Moreover, during the initial phases of SARS-CoV-2 infection, the innate immune response is inhibited by the interaction of nsp1 with EIF3G in ribosomes to break down the host mRNA. It is crucial to remember that the synthesis of viral proteins is unaffected by this breakdown process, which permits the virus to carry on multiplying and spreading throughout the host [24]. HEK293T cells were co-transfected with Myc-N and Flag vectors expressing DDX1, SAMHD1, PCBP2, EIF3G, or DDX3X. The Co-IP assays displayed that the PRRSV nucleocapsid protein interacted specifically with DDX1, SAMHD1, PCBP2, EIF3G, and DDX3X, and no signal was observed with the empty vector (Figure 5). The results indicated that all the proteins that were selected were able to interact with the nucleocapsid protein, which adds a level of confidence to the identification of N-interacting proteins from the LC– MS/MS data.

### 2.6. PRRSV May Inhibit Innate Immunity via Nucleocapsid Protein

Previous studies have reported that PCBP2 negatively regulates the innate immune response to RNA virus infection by regulating the stability of MAVS [25]. In addition, PCBP2 negatively regulates the cGAS-STING pathway through its interaction with cGAS [26]. To assess the influence of PRRSV on host proteins, the mRNA levels of PCBP2 proteins in the PRRSV-infected PAMs were measured by RT-qPCR. The results revealed that the mRNA levels of PCBP2 were increased (Figure 6A). Then, the HEK293T cells were co-transfected with the recombinant vectors Flag-PCBP2 and Myc-N. Western blotting analysis revealed that overexpression of nucleocapsid protein substantially enhanced the quantity of PCBP2 protein (Figure 6B). The results suggest that nucleocapsid protein may inhibit innate immunity by up-regulating PCBP2 expression. These findings indicate that nucleocapsid protein may be a critical protein in suppressing innate immunity during PRRSV infection.

## 3. Discussion

The interaction between the viral and host cell proteins is very important in understanding the PRRSV infection processes and host cellular responses. Among them, the nucleocapsid protein plays a key role in virus proliferation. The nucleocapsid protein is the only component of the viral capsid and recruits nucleolar proteins to promote viral replication [9]. Sagong and Lee [27] found that nucleocapsid proteins control the production of interferon-β by preventing IRF3 from being activated in immortalized porcine alveolar macrophages. It has been reported that the PRRSV nucleocapsid protein regulates the JAK/STAT signaling pathway by upregulating cytokine signaling pathway 1 (SOCS1) [28]. In addition, the nucleocapsid protein activates NF-κB, which promotes inflammation [11]. Subsequent studies have identified that nucleocapsid protein could enhance the activation of NF-kB by interacting with the N-terminal tetramer of DExD/H-box protein 36 (DHX36) [29]. Therefore, identifying host proteins that interact with the nucleocapsid protein can help elucidate the biological processes of PRRSV infection and identify potential drug targets.

Previous studies have reported the screening of the interactions between PRRSV proteins and host proteins; however, the identification of N-interacting host proteins is scarce [1,13,30,31,32,33,34,35]. Jourdan and Osorio [13] identified 65 host proteins interacting with nucleocapsid protein in HEK293T cells. However, the natural target cells of PRRSV infection are PAMs, so the PAMs-based proteomic data is closer to the true infection state. Finally, this study performed immunoprecipitation of specific antibodies using the PRRSV-infected PAMs. Silver staining was then used to reveal the cellular proteins that were bound to the PRRSV nucleocapsid protein. Specific strips were cut for trypsin digestion and LC– MS/MS analysis, resulting in the identification of 349 potential PRRSV N-interacting host proteins when using cutoff values (Figure 1).

To gain a deeper understanding of the identified N-interacting proteins, we conducted bioinformatic analysis to extract pertinent information. The N-interacting proteins’ functional enrichment study showed enrichment in various functional GO categories. Protein stabilization, protein transport, and translation were the top-ranked GO biological processes categories. The proteins showed a high enrichment in binding, catalytic activity, and structural molecular activity for GO molecular functions. Similar results have been reported in previous studies [13,27]. However, our study uncovered unreported associations of the nucleocapsid protein with proteins involved in the negative regulation of apoptotic processes, protein import into nucleus, and heat shock protein binding (Figure 2). In addition, nucleocapsid protein might be connected to unreported processes including ferroptosis and proteasome, according to KEGG pathways enrichment data (Figure 3).

Subsequently, we used STRING analysis to clarify the functional relationships of these N-interacting proteins (Figure 4). The PPI network of nucleocapsid protein and its inter-agents was generated with a confidence of 0.7. PPI analysis showed that N-interacting proteins are divided into different clusters, and there is a strong association between the same cluster proteins. There is a great possibility that proteins in the same cluster can work together to perform their biological functions. Our study thus greatly expands the knowledge of the nucleocapsid protein interaction set and presents testable hypotheses for elucidating new nucleocapsid protein functions.

In this study, we selected five host proteins, DDX1, SAMHD1, PCBP2, EIF3G, and DDX3X, known to regulate innate immunity and viral protein translation, and verified their interactions with the nucleocapsid protein through Co-IP. The results showed that all the identified proteins could interact with the nucleocapsid protein (Figure 5A–D). Validation results for the specific proteins added confidence to the LC– MS/MS data for these potentially N-interacting proteins. Our research significantly enhances our understanding of the nucleocapsid proteins’ new functionalities. It is important to note, however, that these N-interacting proteins may interact with the nucleocapsid protein directly or indirectly through protein complexes. As a result, additional studies will be required for significant confirmation.

PCBP2 is a member of a protein family that binds to the poly(C) stretches in both DNA and RNA [36,37]. It is involved in the regulation of protein–protein interactions [25,38], protein translation [39,40], and mRNA stability [41,42]. Previous research has demonstrated that PCBP2 controls MAVS stability via the HECT ubiquitin ligase AIP4 and negatively modulates the innate immune response to RNA virus infection [25]. In addition, PCBP2 negatively regulates antiviral signaling by specifically interacting with cGAS [26]. Most importantly, PCBP2 can interact with nsp1β and promote PRRSV replication [43]. PCBP2 was also screened in our database and its interaction with the nucleocapsid protein was verified (Figure 5E). Subsequently, we examined the effect of PRRSV infection on PCBP2. The results showed that the PAMs infected with PRRSV could upregulate PCBP2 mRNA level (Figure 6A). Then, it was found that the overexpression of the nucleocapsid protein in HEK293T cells could upregulate the level of PCBP2 protein (Figure 6B). Thus, we hypothesized that the nucleocapsid protein may inhibit innate immunity by interacting with PCBP2 and promoting its expression, as well as facilitate PRRSV replication.

In summary, by analyzing and validating the function of N-interacting proteins, we described the potential regulatory role of the nucleocapsid protein on innate immunity during PRRSV infection. The validation and characterization of these interactions will contribute to the resolution of host–pathogen interaction mechanisms and also provide a theoretical basis for identifying new therapeutic targets to control PRRSV infection.

## 4. Materials and Methods

### 4.1. Cells and Viruses

The MARC-145 cells originated from a monkey kidney cell line that is extremely susceptible to PRRSV infection. For the maintenance of the MARC-145 cells and HEK293T cells, Dulbecco’s Modified Eagle’s Medium (DMEM; Solarbio^®^ Life Sciences) with 10% heat-inactivated fetal bovine serum (FBS; ExCell Bio) was used. As previously reported [44], the PAMs were obtained from healthy six-week-old Large White–Dutch Landrace crossbred piglets using the bronchoalveolar lavage procedure. The PAMs were then housed in RPMI Medium 1640 (Solarbio^®^ Life Sciences) supplemented with 10% fetal bovine serum. Every cell was cultivated at 37 °C in a humidified incubator with 5% CO_2_. In MARC-145 cells, the PRRSV strain HN071 (GenBank accession number KX766378.1) was propagated.

### 4.2. Antibodies and Reagents

HRP-β-actin (HRP-60008) was procured from Proteintech. Anti-Myc rabbit (2272) was procured from CST. Sigma provided the Anti-Flag M2 Mouse mAb (F1804) and Anti-Flag^®^ M2 affinity gel (A2220). The following items were purchased from seracare: KPL Peroxidase-Labeled Antibody To Mouse IgG(H + L) (5220-0341), KPL Peroxidase-Labeled Antibody To Rabbit IgG(H + L) (5220-0336), and FITC-labeled Goat Anti-Mouse IgG(H + L) (5230-0426). The Mouse Anti-PRRSV-N mAb were produced by our lab. The enhanced chemiluminescence (ECL) detection reagent was purchased from Solarbio^®^ Life Sciences (PE0030). Beyotime provided the following products: Protein A + G Agarose (P2055), PMSF Solution (100 mM) (ST507), NP-40 Lysis Buffer (P0013F), and Fast Silver Stain Kit (P0017S). The following products were purchased from Vazyme: ChamQ Universal SYBR qPCR Master Mix (Q711-02), HiScript II Q RT SuperMix for RT-qPCR (+gDNA wiper) (R223-01), and RNA isolater Total RNA Extraction Reagent (R401-01).

### 4.3. Plasmid Construction and Cell Transfection

The nucleocapsid protein gene was cloned from the PRRSV Hn07-1 genome by RT-PCR and was inserted into pCMV-Myc to produce Myc-N. DDX1, SAMHD1, PCBP2, EIF3G, and DDX3X were cloned from the PAMs genome and subsequently constructed into the pCMV-Flag vector. All constructed plasmids were analyzed and verified by DNA sequencing. Appendix A contains a list of all primers and synthesized oligodeoxynucleotide sequences. The HEK293T or MARC-145 cells were planted at an appropriate density in predetermined plates. The specified recombinant vectors were cotransfected into cells using Lipofectamine 2000 reagent (Thermo Fisher Scientific) in accordance with the manufacturer’s instructions once the cells had achieved 80–85% confluence.

### 4.4. RNA Isolation and RT-qPCR

PAMs were inoculated into 12-well plates and infected with PRRSV. Cell samples were harvested 36 h after infection. Total RNA was extracted using the Super FastPure Cell RNA Isolation Kit (Vazyme) according to the manufacturer’s instructions, and the purity and concentration of RNA were measured with a spectrophotometer. Hiscript III Reverse Transcriptase (Vazyme) was used to generate cDNA, and ChamQ Universal SYBR qPCR Master Mix (Vazyme) was used to carry out the qPCR. A single cycle of denaturation at 95 °C for 30 s was followed by 40 cycles of amplification at 95 °C for 5 s and 60 °C for 34 s. A final melting cycle was added to construct a melting curve and check the product specificity. The specificities of the PCR products were confirmed by a single peak found in the melting curve. Primers are shown in Appendix A. As an internal control, β-actin RNA was employed. Using the 2^−∆∆Ct^ method, the relative quantification of the target gene was determined.

### 4.5. Western Blotting

Total proteins were extracted from the cell samples by lysing them in lysis buffer (5% SDS, 1% Triton X-100, 50 mM Tris-HCl, 150 mM NaCl) and boiling them for 10 min at 100 °C in 4× loading buffer. Following SDS-PAGE separation, the protein samples were transferred onto NC membranes. The membranes were blocked for 1 h at 37 °C with 5% nonfat dry milk with 0.1% Tween 20 (Solar-bio, T8220). Next, the membranes were treated for 8 h at 4 °C with primary antibodies, and then for 1 h at 37 °C with HRP-labeled anti-rabbit/mouse IgG. An extremely sensitive ECL Chemiluminescence Detection Kit was then used to detect the signals.

### 4.6. Co-IP

HEK293T cells were grown in 6-well plates to 80% confluency and then co-transfected with multiple plasmids using Lipofectamine^®^ 2000 Transfection Reagent. NP-40 was used to lyse the transfected cells after 36 h. Approximately 10 μL of anti-Flag agarose beads were incubated with 0.5 mL of the lysate of each sample for 4 h at 4 °C. Then, 0.8 mL of PBS was used to wash the sepharose beads five times. Western blotting was used to evaluate the precipitates.

### 4.7. LC– MS/MS

The silver staining gels from teh Co-IP experiments were mixed and subjected to protein identification by LC– MS/MS analysis in APTBio (Shanghai, China). In a nutshell, the gel fragments were freeze-dried after being dyed with 30% acetonitrile and 100 mM NH_4_HCO_3_. The gel fragments were alkylated with 200 mM iodoacetamide (in the dark, 25 °C, 20 min) after being reduced with 100 mM DTT (56 °C, 30 min). The gels were treated with 100 mM NH_4_HCO_3_, shrunk with acetonitrile once more, and then incubated for 20 h at 37 °C with trypsin (2.5–10 ng/μL). Using 0.1% TFA and 60% acetonitrile, the peptides were extracted. A nano-flow HPLC (LTQ VELOS, Thermo Finnigan, San Jose, CA, USA) was used to separate the peptides.

### 4.8. Bioinformatics Analysis

All N-interacting proteins were subjected to GO analysis and KEGG pathway annotation using DAVID Bioinformatics Resources (http://david.ncifcrf.gov, accessed on 6 February 2024). The N-interacting proteins were analyzed for enrichment in GO and KEGG pathway annotations using Fisher’s exact test (*p* < 0.05). Furthermore, the interaction of target proteins was established using the STRING database (https://cn.string-db.org, accessed on 10 March 2024).

### 4.9. Statistics and Data Analysis

GraphPad Prism (Version 7) was used to analyze all the data, which were then presented as mean ± SD. *t*-tests were used to compare all groups. *P*-values of less than 0.05 were used to determine statistical significance of differences.

## Figures and Tables

**Figure 1 ijms-25-07219-f001:**
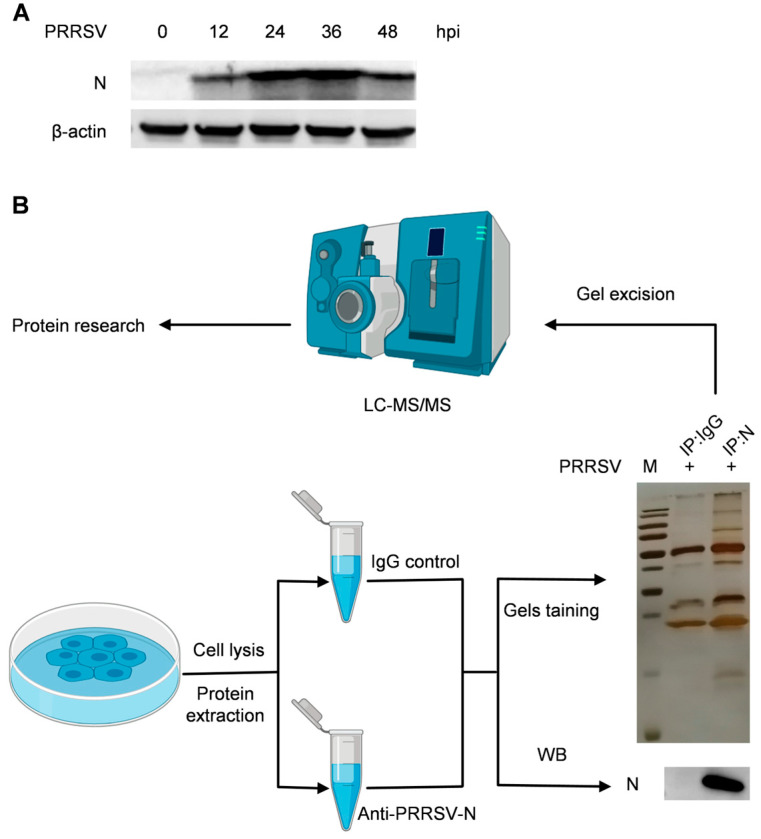
The cellular proteins that interact with the PRRSV nucleocapsid protein are identified using Co-IP experiments. (**A**) Western blotting was performed on cell lysates from PRRSV-infected PAMs at various time points using anti-PRRSV-N and anti-β-actin mAbs. (**B**) PAMs were infected with PRRSV at a multiplicity of infection (MOI) of 1 for 36 h and then collected for Co-IP. The resulting cell lysates were subjected to immunoblotting with the indicated antibody. Meanwhile, following SDS-PAGE electrophoresis, the gel was revealed using silver staining, and the gel fragment was cut for LC– MS/MS analysis.

**Figure 2 ijms-25-07219-f002:**
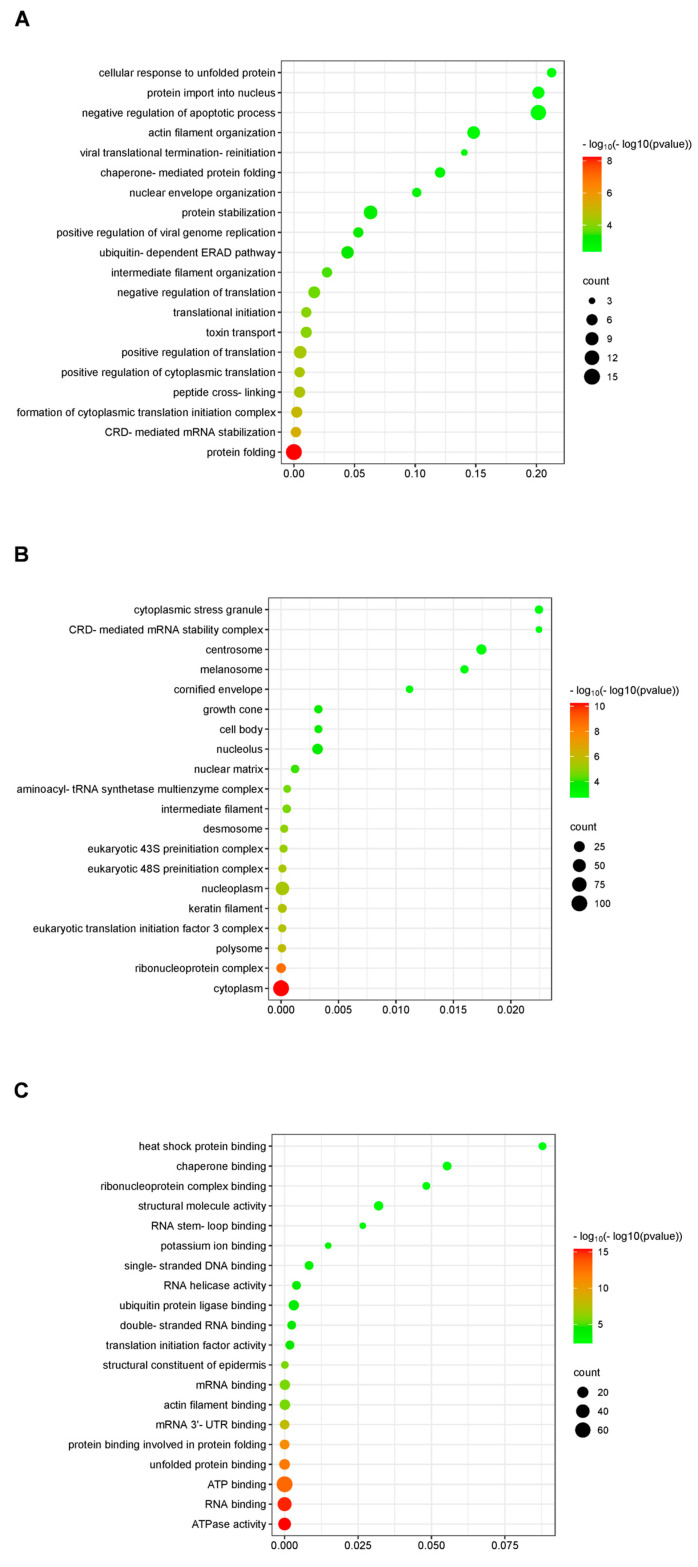
The GO database’s annotation of proteins involved in interactions with PRRSV nucleocapsid protein. (**A**) Biological processes; (**B**) Cellular components; and (**C**) Molecular function. The top 20 enriched terms were revealed. The enrichment is represented by the figure’s abscissa. The hyper-geometric test’s *p* value is shown by the color of the dot. The color spectrum includes red and green. The rating decreases with increasing redness in the color. The quantity of proteins in the associated route is indicated by the size of the point.

**Figure 3 ijms-25-07219-f003:**
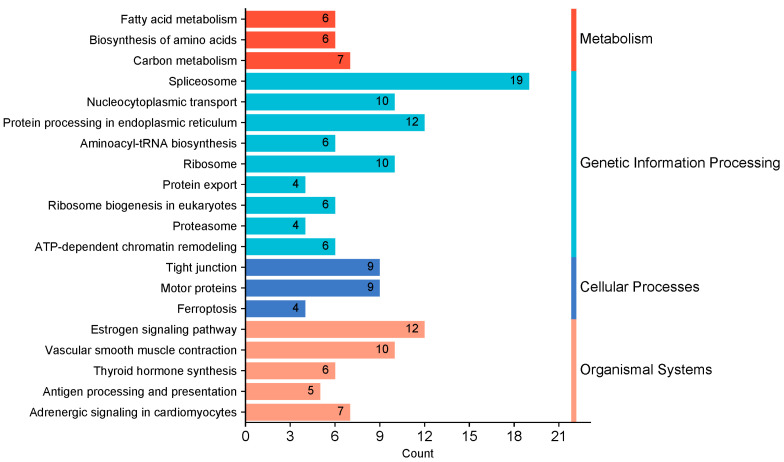
The graph depicts the enriched pathways targeted by N-interacting proteins as evaluated by the KEGG functional annotation pathway database.

**Figure 4 ijms-25-07219-f004:**
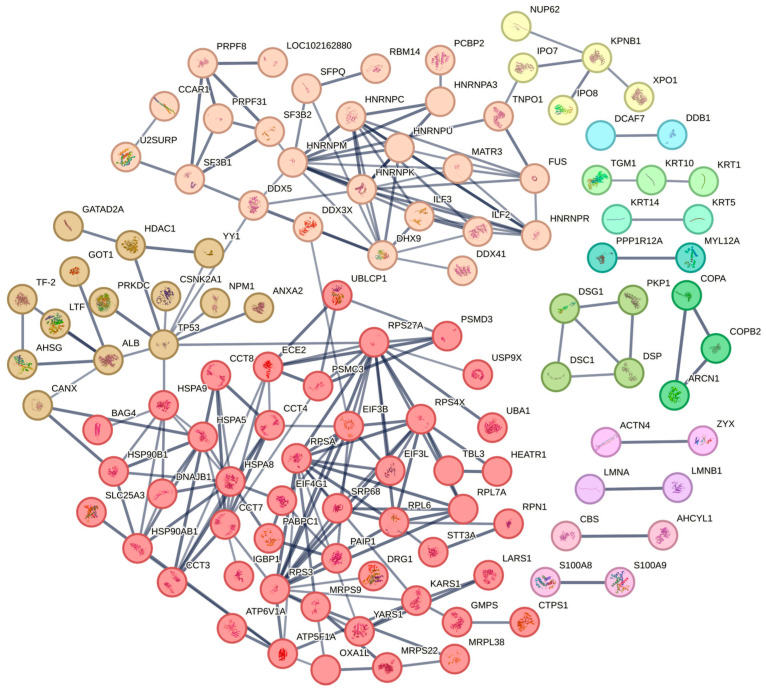
PPI network study of N-interacting proteins. The nucleocapsid protein interaction network was generated from the STRING database with a confidence value above 0.7. Interacting proteins with the same color were classified into one cluster.

**Figure 5 ijms-25-07219-f005:**
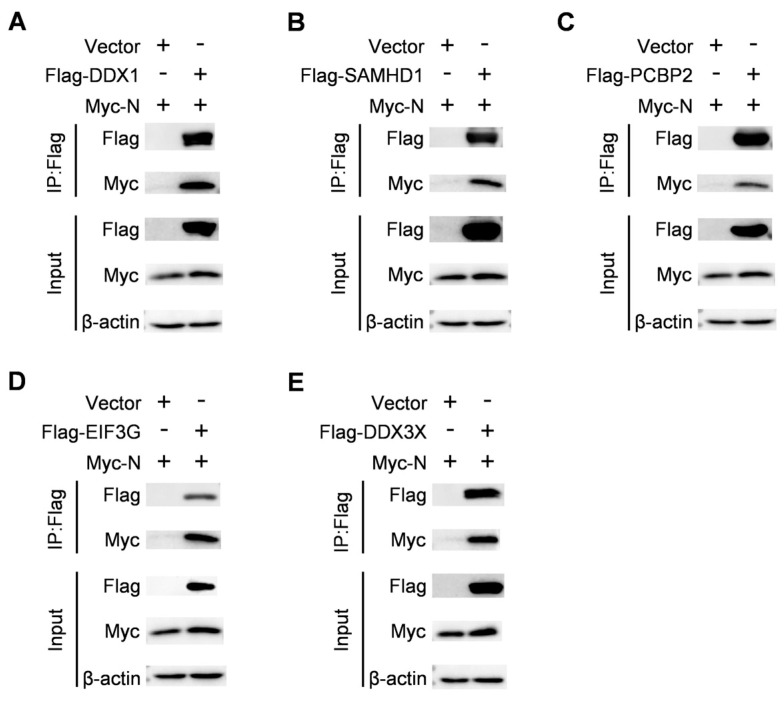
Identifying the chosen proteins that interact with the nucleocapsid protein. HEK293T cells were co-transfected with Flag vectors expressing the specified host proteins ((**A**): DDX1, (**B**): SAMHD1, (**C**): PCBP2, (**D**): EIF3G, (**E**): DDX3X) and Myc-N for 24 h. Anti-Flag IgG beads were then used to immunoprecipitate the cell lysates, and the corresponding antibodies were subsequently used for immunoblotting.

**Figure 6 ijms-25-07219-f006:**
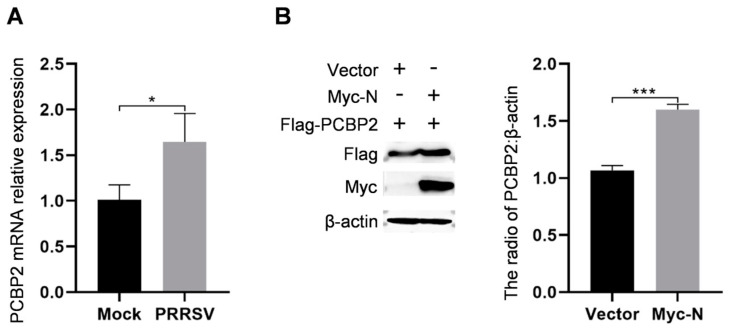
The influence of PRRSV and nucleocapsid protein on PCBP2 expression levels. (**A**) The PAMs were infected with PRRSV at a MOI of one for 36 h. The cells were collected, and then the mRNA levels of PCBP2 were detected using RT-qPCR analysis. (**B**) HEK293T cells were co-transfected with Flag-PCBP2 and Myc-N for 24 h. The resulting cell lysates were immunoblotted with the specified antibody. ImageJ software was used to calculate the relative quantification of protein bands. Error bars represent means ± SD from three separate tests. Student’s *t*-test: * *p* < 0.05; *** *p* < 0.001 against control.

## Data Availability

All data are included in the manuscript and Appendix A. The data presented in this research have been deposited to the OMIX, China National Center for Bioinformation/Beijing Institute of Genomics, Chinese Academy of Sciences [https://ngdc.cncb.ac.cn/omix: accession no. OMIX006385 (Co-IP-LC-MS/MS), accessed on 20 May 2024].

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
