# Peer review of "Mass Spectrometry-Based Proteomic Analysis of Potential Host Proteins Interacting with N in PRRSV-Infected PAMs"

_ijms, 2024, doi:10.3390/ijms25137219_

Round 1
Reviewer 1 Report
Comments and Suggestions for Authors
The authors used a proteomics-based approach to study the potential host proteins interacting with PRRSV nucleocapsid protein in the infected PAMs and studied the protein-protein interaction network. Several points need to be addressed before this manuscript can be considered for the next step:
1. Please don’t use N, instead, change N throughout the manuscript to nucleocapsid protein.
2. Line 82, what is the negative control “mock” infected PAM? Please provide more details and clarify.
3. Section 2.4, the PPI network analysis is not well analyzed. What are the findings? Are the different colored clusters associated with different pathways? I don’t see how the KEGG pathway information was derived based on the manuscript and data presented.
4. In the Discussion section, the authors used long paragraphs to talk about the host proteins selected for CO-IP, the majority of this introductory information should be moved to section 2.5.
Comments on the Quality of English LanguageGenerally good, only minor issues observed.
For example, in line 126 "metabolism" instead of "imetabolism"
Author Response
Dear Reviewer:
Thank you very much for your comments and professional advice for our manuscript, entitled “Mass spectrometry-based proteomic analysis of potential host proteins interacting with N in PRRSV-infected PAMs” (Manuscript ID: ijms-3049532). These opinions help to improve academic rigor of our article. Based on your suggestions and requests, we have modified the manuscript using comment mode. At the same time, the language description problem in the paper is optimized. Furthermore, we would like to show the details as follows:
Comment 1: Please don’t use N, instead, change N throughout the manuscript to nucleocapsid protein.
Response 1: Thanks for your reminding, we have changed the N in the manuscript to nucleocapsid protein.
Comment 2: Line 82, what is the negative control “mock” infected PAM? Please provide more details and clarify.
Response 2: Thank you for your question. Sorry for our unclear description. We give a detailed description of the negative control “mock” infection with PAM, in lines 86-89 of the manuscript.
Comment 3: Section 2.4, the PPI network analysis is not well analyzed. What are the findings? Are the different colored clusters associated with different pathways? I don’t see how the KEGG pathway information was derived based on the manuscript and data presented.
Response 3: Thank you for your questions, and Section 2.4 is dedicated to further understanding the functional relationships between N-interacting proteins. Different colors represent different clusters, and there is a strong correlation between proteins in the same cluster, but different clusters do not represent different pathways. In addition, different clusters of proteins come together to perform their biological functions in concert to a large extent. Sorry for our earlier vague description, we have revised the relevant content, including lines 151-159 and 258-265.
Comment 4: In the Discussion section, the authors used long paragraphs to talk about the host proteins selected for CO-IP, the majority of this introductory information should be moved to section 2.5.
Response 4: At your suggestion, we have moved the section of the discussion on host protein selection to lines 164-184 of section 2.5.

Reviewer 2 Report
Comments and Suggestions for Authors
The manuscript entitled “Mass spectrometry-based proteomic analysis of potential host proteins interacting with N in PRRSV-infected PAMs” explores the interaction between PRRSV N protein and host proteins. Therefore, their results suggest that N protein interacts with several immune related proteins in pig, especially ATP-dependent RNA helicase, Sterile alpha motif 66 and HD domain-containing protein 1, Poly(rC)-binding protein 2, Eukaryotic translation initiation factor 3 subunit G. Although the data looks interesting, I think there is a need for more data to support the hypothesis that N is modulation the immune response in the host cells.
Major comments:
1- It is not clear why the authors chose those specific host protein to work with? In the discussion they mentioned that they chose the targets because they play a role in the immune system but that was not convincing.
2- They are speculation that PRRSV may inhibit the innate immunity via N (which interacts with PCBP2), did the author measure any marker of the Innate immune system? How can they point out N as the main protein that modulates this immune response? Since the PAM were infected with PRRSV, it could happen that other PRRSV protein may be acting or interacting the host proteins. There is little data to support that, also what about the other host protein that N interacts with (for example DDX1, SAMHD1, EIF3G and DDX3X)
Author Response
Dear Reviewer:
Thank you for your comments on our manuscript, entitled “Mass spectrometry-based proteomic analysis of potential host proteins interacting with N in PRRSV-infected PAMs” (Manuscript ID: ijms-3049532). They are all valuable and very helpful for revising and improving our paper, as well as the important guiding significance to our research. Based on your comments, the manuscript was revised using the annotation mode. The point-by-point responses to your comments are as follows:
Comment 1: It is not clear why the authors chose those specific host protein to work with? In the discussion they mentioned that they chose the targets because they play a role in the immune system but that was not convincing.
Response 1: Thank you for your question. In the discussion, we mentioned that N protein plays an important role in the immune system [1-4], and further enriched the introduction of related studies in lines 223-227 of the manuscript. Therefore, by reviewing relevant literature, we selected some target proteins (DDX1, SAMHD1, PCBP2, EIF3G and DDX3X) [5-16] related to innate immunity for verification.
Comment 2: They are speculation that PRRSV may inhibit the innate immunity via N (which interacts with PCBP2), did the author measure any marker of the Innate immune system? How can they point out N as the main protein that modulates this immune response? Since the PAM were infected with PRRSV, it could happen that other PRRSV protein may be acting or interacting the host proteins. There is little data to support that, also what about the other host protein that N interacts with (for example DDX1, SAMHD1, EIF3G and DDX3X).
Response 2: Thank you for your questions. Previous studies have reported that PCBP2 can promote the development of innate immunity[15, 16]. Our results show that N protein can interact with PCBP2 and inhibit its expression. Therefore, we speculate that PRRSV may inhibit innate immunity through N. Moreover, subsequent studies will focus on examining the markers of on innate immune system via N interacting PCBP2. The main content of our paper is the identification of host proteins that interact with N proteins and the regulatory role of proteins associated with innate immunity. In this study, host proteins enriched with anti-PRRSV-N mAb was used, so the detected host proteins all interact with N. As to whether these host proteins interact with other PRRV proteins, we have other people doing that in our lab[17].
References
- Luo, R., L. Fang, Y. Jiang, H. Jin, Y. Wang, D. Wang, H. Chen, and S. Xiao. "Activation of Nf-Κb by Nucleocapsid Protein of the Porcine Reproductive and Respiratory Syndrome Virus." Virus Genes42, no. 1 (2011): 76-81.
- Sagong, M., and C. Lee. "Porcine Reproductive and Respiratory Syndrome Virus Nucleocapsid Protein Modulates Interferon-Β Production by Inhibiting Irf3 Activation in Immortalized Porcine Alveolar Macrophages." Arch Virol156, no. 12 (2011): 2187-95.
- Luo, X., X. X. Chen, S. Qiao, R. Li, S. Xie, X. Zhou, R. Deng, E. M. Zhou, and G. Zhang. "Porcine Reproductive and Respiratory Syndrome Virus Enhances Self-Replication Via Ap-1-Dependent Induction of Socs1." J Immunol204, no. 2 (2020): 394-407.
- Jing, H., Y. Zhou, L. Fang, Z. Ding, D. Wang, W. Ke, H. Chen, and S. Xiao. "Dexd/H-Box Helicase 36 Signaling Via Myeloid Differentiation Primary Response Gene 88 Contributes to Nf-Κb Activation to Type 2 Porcine Reproductive and Respiratory Syndrome Virus Infection." Front Immunol8 (2017): 1365.
- Bonaventure, B., and C. Goujon. "Dexh/D-Box Helicases at the Frontline of Intrinsic and Innate Immunity against Viral Infections." J Gen Virol103, no. 8 (2022).
- Zhang, Z., T. Kim, M. Bao, V. Facchinetti, S. Y. Jung, A. A. Ghaffari, J. Qin, G. Cheng, and Y. J. Liu. "Ddx1, Ddx21, and Dhx36 Helicases Form a Complex with the Adaptor Molecule Trif to Sense Dsrna in Dendritic Cells." Immunity34, no. 6 (2011): 866-78.
- Oshiumi, H., K. Sakai, M. Matsumoto, and T. Seya. "Dead/H Box 3 (Ddx3) Helicase Binds the Rig-I Adaptor Ips-1 to up-Regulate Ifn-Beta-Inducing Potential." Eur J Immunol40, no. 4 (2010): 940-8.
- Cargill, M., R. Venkataraman, and S. Lee. "Dead-Box Rna Helicases and Genome Stability." Genes (Basel)12, no. 10 (2021).
- Goldstone, D. C., V. Ennis-Adeniran, J. J. Hedden, H. C. Groom, G. I. Rice, E. Christodoulou, P. A. Walker, G. Kelly, L. F. Haire, M. W. Yap, L. P. de Carvalho, J. P. Stoye, Y. J. Crow, I. A. Taylor, and M. Webb. "Hiv-1 Restriction Factor Samhd1 Is a Deoxynucleoside Triphosphate Triphosphohydrolase." Nature480, no. 7377 (2011): 379-82.
- Powell, R. D., P. J. Holland, T. Hollis, and F. W. Perrino. "Aicardi-Goutieres Syndrome Gene and Hiv-1 Restriction Factor Samhd1 Is a Dgtp-Regulated Deoxynucleotide Triphosphohydrolase." J Biol Chem286, no. 51 (2011): 43596-600.
- Chen, S., S. Bonifati, Z. Qin, C. St Gelais, K. M. Kodigepalli, B. S. Barrett, S. H. Kim, J. M. Antonucci, K. J. Ladner, O. Buzovetsky, K. M. Knecht, Y. Xiong, J. S. Yount, D. C. Guttridge, M. L. Santiago, and L. Wu. "Samhd1 Suppresses Innate Immune Responses to Viral Infections and Inflammatory Stimuli by Inhibiting the Nf-Κb and Interferon Pathways." Proc Natl Acad Sci U S A115, no. 16 (2018): E3798-e807.
- Espada, C. E., C. St Gelais, S. Bonifati, V. V. Maksimova, M. P. Cahill, S. H. Kim, and L. Wu. "Traf6 and Tak1 Contribute to Samhd1-Mediated Negative Regulation of Nf-Κb Signaling." J Virol95, no. 3 (2021).
- García-de-Gracia, F., A. Gaete-Argel, S. Riquelme-Barrios, C. Pereira-Montecinos, B. Rojas-Araya, P. Aguilera, A. Oyarzún-Arrau, C. Rojas-Fuentes, M. L. Acevedo, J. Chnaiderman, F. Valiente-Echeverría, D. Toro-Ascuy, and R. Soto-Rifo. "Cbp80/20-Dependent Translation Initiation Factor (Ctif) Inhibits Hiv-1 Gag Synthesis by Targeting the Function of the Viral Protein Rev." RNA Biol18, no. 5 (2021): 745-58.
- Abaeva, I. S., Y. Arhab, A. Miścicka, C. U. T. Hellen, and T. V. Pestova. "In Vitro Reconstitution of Sars-Cov-2 Nsp1-Induced Mrna Cleavage Reveals the Key Roles of the N-Terminal Domain of Nsp1 and the Rrm Domain of Eif3g." Genes Dev37, no. 17-18 (2023): 844-60.
- You, F., H. Sun, X. Zhou, W. Sun, S. Liang, Z. Zhai, and Z. Jiang. "Pcbp2 Mediates Degradation of the Adaptor Mavs Via the Hect Ubiquitin Ligase Aip4." Nat Immunol10, no. 12 (2009): 1300-8.
- Gu, H., J. Yang, J. Zhang, Y. Song, Y. Zhang, P. Xu, Y. Zhu, L. Wang, P. Zhang, L. Li, D. Chen, and Q. Sun. "Pcbp2 Maintains Antiviral Signaling Homeostasis by Regulating Cgas Enzymatic Activity Via Antagonizing Its Condensation." Nat Commun13, no. 1 (2022): 1564.
- Li, W., Y. Wang, M. Zhang, S. Zhao, M. Wang, R. Zhao, J. Chen, Y. Zhang, and P. Xia. "Mass Spectrometry-Based Proteomic Analysis of Potential Host Proteins Interacting with Gp5 in Prrsv-Infected Pams." Int J Mol Sci25, no. 5 (2024).

Round 2
Reviewer 2 Report
Comments and Suggestions for Authors
Thanks for the replies.